# Chloroplast Thylakoidal Ascorbate Peroxidase, PtotAPX, Has Enhanced Resistance to Oxidative Stress in *Populus tomentosa*

**DOI:** 10.3390/ijms23063340

**Published:** 2022-03-19

**Authors:** Conghui Li, Jiaxin Li, Xihua Du, Jiaxue Zhang, Yirong Zou, Yadi Liu, Ying Li, Hongyan Lin, Hui Li, Di Liu, Hai Lu

**Affiliations:** 1Beijing Advanced Innovation Center for Tree Breeding by Molecular Design, Beijing Forestry University, Beijing 100083, China; liconghui0225@163.com (C.L.); koni90@163.com (J.Z.); zouyr706@163.com (Y.Z.); lihui830@bjfu.edu.cn (H.L.); 2The Tree and Ornamental Plant Breeding and Biotechnology Laboratory of National Forestry and Grassland Administration, College of Biological Sciences and Biotechnology, Beijing Forestry University, Beijing 100083, China; lijiaxin123ing@163.com (J.L.); shengkeliuyadi@126.com (Y.L.); yingyingyaojiayou@163.com (Y.L.); linhongyan1116@163.com (H.L.); 3Shandong Provincial Key Laboratory of Plant Stress Research, College of Life Sciences, Shandong Normal University, Jinan 250014, China; duxihua@163.com

**Keywords:** *Populus tomentosa*, chloroplast thylakoidal ascorbate peroxidase, reactive oxygen species

## Abstract

Chloroplasts are the most major producers of reactive oxygen species (ROS) during photosynthesis. However, the function of thylakoid ascorbate peroxidase (tAPX) in response to oxidative stress in wood trees is largely unknown. Our results showed that PtotAPX of *Populus tomentosa* could effectively utilize ascorbic acid (AsA) to hydrolyze hydrogen peroxide (H_2_O_2_) *in vitro*. The overexpression or antisense of *PtotAPX* (OX-PtotAPX or anti-PtotAPX, respectively) in *Populus tomentosa* plants did not significantly affect plant morphology during plant growth. When treated with methyl viologen (MV), the OX-PtotAPX plants exhibited less morphological damage under stress conditions compared to WT plants. OX-PtotAPX plants maintained lower H_2_O_2_ levels and malondialdehyde (MDA) contents, but more reduced AsA levels, a higher photosynthetic rate (Pn), and the maximal photochemical efficiency of PSII (Fv/Fm), whereas anti-PtotAPX plants showed the opposite phenotype. Furthermore, the activity of APX was slightly higher in OX-PtotAPX under normal growth conditions, and this activity significantly decreased after stress treatment, which was the lowest in anti-P. Based on these results, we propose that *PtotAPX* is important for protecting the photosynthetic machinery under severe oxidative stress conditions in *P. tomentosa*, and is a potential genetic resource for regulating the stress tolerance of woody plants.

## 1. Introduction

Reactive oxygen species (ROS) play a central role in many signaling pathways involved in stress perception, photosynthesis regulation, programmed cell death, as well as plant growth and development [1]. However, high concentrations of H_2_O_2_ can cause membrane lipid peroxidation, protein oxidation, and DNA and RNA damage, resulting in damage to cell structure, and eventually cell death [2]. Plant antioxidative enzymes, such as catalase (CAT), superoxide dismutase (SOD), glutathione S-transferase (GST), and ascorbate peroxidase (APX), regulate the rates of ROS production and scavenging in different cellular compartments, thereby maintaining the steady-state level of ROS [3].

Chloroplasts are the main site of ROS production in plant cells. Therefore, chloroplastic APX (chlAPX), especially thylakoid APX (tAPX), is directly involved in the water–water cycle (WWC, which can scavenge ROS in chloroplasts) based on its high susceptibility to H_2_O_2_ and plays a major role in scavenging the H_2_O_2_ generated by green tissues during photosynthesis [3,4]. However, under low AsA levels, chlAPX was extremely sensitive to H_2_O_2_ and rapidly inactivated [5]. Under photo-oxidative stress conditions, a significant decrease in the redox status of AsA was observed in wild-type tobacco [6]. A high level of AsA is essential for maintaining the antioxidant system. Maintaining the ROS levels within an appropriate range promotes plant health, whereas ROS levels that are too high and exceed the threshold boundary impair plant growth and development [7]. Thus, the detoxification of H_2_O_2_ is necessary to avoid damage to the photosynthetic mechanisms and plant growth. However, researchers failed to find a stress-sensitive phenotype among loss-of-function mutants under normal conditions [8]. For example, *Arabidopsis* or rice chlAPX mutants exhibit a normal phenotype and biochemical and physiological attributes under normal growth conditions [3]. Several studies have suggested that cross-compartment protection is present in plant cells for the removal of ROS in the cytosol, chloroplasts, peroxisomes, and mitochondria [3,9]. However, the extent of cross-compartment protection may be limited, especially under serious stress conditions. Previous studies showed that, under severe stress conditions, the *KO-s/mAPX Arabidopsis* mutant was sensitive to paraquat (Pq) at concentrations higher than 50 mM [3]. Arabidopsis tAPX antisense seedlings are more sensitive to Pq-induced oxidative damage, whereas the overexpression of tAPX increases Pq tolerance [10]. In addition, *Arabidopsis* knockout mutants (*KO-tAPX* and *KO-sAPX*), particularly *KO-tAPX*, accumulated higher levels of H_2_O_2_ and oxidized proteins in both mutant plants than in WT plants when treated with methyl viologen (MV) under light [11]. Therefore, the amount of H_2_O_2_ and over-accumulation of ROS may trigger plant damage or even cell death.

The cell wall plays a key role in the interactions between cells and the environment to protect the cell from environmental challenges. Interestingly, H_2_O_2_ acts as a developmental signal in regulating fiber development by controlling its levels in secondary cell wall synthesis [12]. Low levels of H_2_O_2_ help in secondary cell wall polymerization as a signal and promote cell expansion or enlargement through their involvement in plant cell wall loosening [13]. During abiotic stress exacerbations, the plant cell wall acts as the first line of defense against environmental effects to protect plant growth. Cell-wall lignification occurs as a stress response and provides resistance to plant tissues. Previous studies have reported that APX members are responsible for reactive oxygen species homeostasis in secondary cell wall biosynthesis during cotton fiber development stages [14]. Moreover, cytosolic APXs also improve redox homeostasis during rapid fiber elongation. Under normal conditions in cotton, the overexpression or specific downregulation of *GhAPX1AT/DT* showed no obvious change, but a higher tolerance to oxidative stress was observed in the overexpression lines [12].

Methyl viologen (MV), the main ingredient of Paraquat, transferred photosystem I electrons to O_2_, causing oxidative damage in cells. When plants were exposed to sunlight, high levels of ROS were generated in chloroplasts. It was then rapidly oxidized by oxygen molecules, and the oxygen was reduced to superoxide, O_2_^−^, and finally, highly toxic OH radicals. Currently, oxidative stress is commonly used as an endogenous inducer with methyl viologen. *Populus tomentosa* is one of the most adaptable tree species in the world, and suitable for planting in regions affected by drought and high salinity. A better understanding of the role if chlAPXs in woody plants could contribute to a more complete understanding of the molecular and physiological mechanisms that respond to abiotic stress. Thus, we used MV as an endogenous inducer to study oxidative damage to plants.

In order to better clarify the physiological function of the tAPX regulating of ROS levels, in previous studies, we generated overexpressing or antisense *PtotAPX* transgenic *P. tomentosa* lines [15]. Additionally, in this study, we investigated the physiological function of chloroplast *PtotAPX* in the regulation of H_2_O_2_ levels and protection against oxidative stress in woody plants using *P. tomentosa* as a model plant. Upon the induction of oxidative stress by MV treatment, we found that wild-type plants suffered more damage than the untreated plants at the morphological, molecular, and cellular levels. Moreover, in contrast to the *PtotAPX* overexpression plants, the antisense *PtotAPX* plants were more damaged under stress conditions compared to WT plants; however, there was no significant difference between WT and transgenic plants under untreated conditions. These findings suggest that *PtotAPX* is important for the protection of photosynthetic mechanisms under severe oxidative stress conditions.

## 2. Results

### 2.1. Upregulation or Downregulation of PtotAPX in Transgenic Populus tomentosa Plants Results in Altered Plant Growth

To determine the function of chloroplast thylakiod APX (*PtotAPX*) in *Populus*, overexpression-*PtotAPX* (OX-P) or antisense-*PtotAPX* (anti-P) transgenic plant lines obtained in previous studies were subjected to further analyses. During plant growth, no significant differences in plant height and leaf size were observed between transgenic and WT plants, and the shape of leaves was also similar (Figure 1A,B). qRT-PCR showed that the transcript level of *PtotAPX* increased to 239% in the OX-P plants and decreased to 0.05% in anti-P plants compared to wild-type plants (WT; Figure 1C). TEM showed that the chloroplasts appeared to be arranged as an oval on the cell wall side, with a clear and intact bilayer membrane structure under normal growth conditions. Thylakoids displayed an elaborate architecture and narrow grana-stroma in the wild-type plants, parallel to the longitudinal axis of the plastids (Figure 1D,G). Similar results were observed for the structures of chloroplasts in OX-P (Figure 1E,H) or anti-P (Figure 1F,I) plants, with a clear and orderly arrangement of thylakoid lamellar structures.

To further investigate if the physiological activity changed during plant growth, the PSII maximum photochemical efficiency (Fv/Fm) and PSII quantum yield were monitored under normal growth conditions using a pulse-amplitude-modulated (PAM) fluorimeter (Figure 1J–O). The spatial distribution of Fv/Fm values is represented with a color scale ranging from black (Fv/Fm = 0.0) to purple (Fv/Fm = 1.0) with red, orange, yellow, green, blue, and dark blue to violet in between. The color of Fv/Fm and Yiled among WT and transgenic plants had no significant change except for the PSII quantum yield of OX-P. These results showed that the photosynthetic activity of anti-P, OX-P, and the WT showed little difference (Figure 1J–O).

The spatial distribution of Fv/Fm values and yield ranged from black (0.0) to purple (1.0). H_2_O_2_ can participate as a substrate in the polymerization of lignin [16]. To further clarify whether the changes in the levels of H_2_O_2_ affected the development of xylem, paraffin section and TEM analyses were performed on the stems between wild-type and transgenic plants. There were no significant differences in the cell morphology of the xylem or the xylem thickness between transgenic and wild-type plants in paraffin sections (Figure 2A–C,G). In addition, phloem thickness was also measured between transgenic and wild-type plants in paraffin sections, showing no significant differences in statistics (Figure 2A–C,H). To further examine the differences in xylem cell morphology, TEM was performed, and the fiber cell wall thickness and fiber cell size were measured using Image J software (Figure 2D–F,H–I). Our results showed that the thickness of the fiber cell wall of anti-P plants decreased by 46.63%, compared with that of WT, whereas there were no significant differences in the fiber cell wall thickness of OX-P plants. However, the fiber cell size of transgenic lines with OX-P or anti-P plants did not show any obvious changes (Figure 2J). The decrease in the fiber cell wall thickness in anti-P plants, although statistically significant, did not affect plant growth.

### 2.2. Decreased PtotAPX Content in Antisense Transgenic Plants Leads to More Chloroplastic Structure Damage in MV-Treated Leaves

Chloroplasts are extremely sensitive to external H_2_O_2_ applications. To further verify the protective effect of *PtotAPX* on chloroplast structure damage, we treated the leaves of transgenic *P. tomentosa* plants with 100 μmol·L^−1^ MV for 5 h. Wild-type, OX-*PtotAPX*, and anti-*PtotAPX* transgenic plant lines treated with MV are referred to as WT-M, OX-P-M, and anti-P-M, respectively. As shown in Figure 3, anti-P leaves were severely damaged after MV treatment, showing more transparent spots compared to the WT, while the leaves were intact in OX-P, suggesting that *PtotAPX* is essential for protecting the chloroplast.

To further determine whether the subcellular structure of chloroplasts was altered under oxidative stress, the cellular ultrastructure of chloroplasts was investigated using TEM (Figure 3D–I). After a 5 h oxidative stress treatment, compared with the untreated control, the structure of chloroplasts of WT plants under oxidative stress suffered significantly more damage (Figure 3D,G). The thylakoid lamellar structure began to become loose and disordered, and the structure of the chloroplasts seemed to be more globular. The distance between the pairs of grana membranes was 20.97 ± 1.56 nm, which was larger than that of the WT under untreated conditions (9.07 ± 1.01 nm). However, the distance between the pairs of grana membranes had only a slight increase in OX-P with MV-treated (9.33 ± 1.35 nm) compared with that untreated with MV (9.77 ± 1.63 nm; Appendix A). Compared with MV-treated wild-type plants, the structure of the thylakoid lamellar and cell membrane system of anti-P was too fuzzy to be distinguished, and some cells were lysed (Figure 3E,H). Therefore, the distance between the pairs of grana membranes was not detected (Appendix A). In contrast, the chloroplasts in OX-P plants under oxidative stress exhibited a regular cell shape and a typical chloroplast appearance (Figure 3E,H). In addition, more abundant and larger PGs appeared in anti-P plants under stress (Figure 3F,I).

To assess oxidative resistance, the Pn and Fv/Fm of 6-month-old transgenic and wild type plants, transplanted from culture, were measured after treatment with 100 μmol·L^−1^ MV for 24 h under room conditions. Pn and Fv/Fm decreased markedly after MV treatment. Compared with untreated wild-type plants, Pn was decreased by 41.91%, 21.80%, and 57.97% of WT, OX-P, and anti-P under oxidative stress, respectively (Figure 3B). The photosynthetic rate decreased more in anti-P than in WT after MV treatment. Consistently, the maximal photochemical efficiency of PSII in WT and transgenic plants under oxidative stress changed significantly compared with the control condition. The Fv/Fm rates of WT, OX-P, and anti-P under oxidative stress were decreased by 25.71%, 15.22%, and 41.86%, respectively (Figure 3C). The increased maximum efficiency of PSII was detected in OX-P compared to WT after treatment with MV and was found to decrease in anti-P (Figure 3J). Similar results of PSII quantum yield were detected in transgenic plants with increased OX-P compared to WT, but decreased in anti-P (Figure 3J). Meanwhile, the leaves of anti-P showed a large dark necrosis area, showing a more serious stress damage, and conversely, no obvious damage showed in OX-P, which was consistent with the Pn and Fv/Fm data (Figure 3J). In summary, these results indicate that the *PtotAPX* gene can prevent oxidative stress damage to plants and protect the chloroplast structure after chloroplast maturation, thereby maintaining chloroplast function and normal physiological metabolism of the plants.

### 2.3. Decreased PtotAPX Content in Antisense Transgenic Plants Leads to the Change in Oxidant or Reduced Content in MV-Treated Leaves, Resulted in Chloroplast Structure Damage

To further investigate if the physiological index changed under normal growth conditions, the photosynthetic rate (Pn) and maximal photochemical efficiency of PSII (Fv/Fm) were determined (Figure 3B,C). No significant differences were observed in Fv/Fm, while the Pn in OX-P was slightly higher than that in the WT. The accumulation of H_2_O_2_ in anti-P plants was higher than that in WT plants, while the accumulation of H_2_O_2_ in OX-P plants was not significantly different from that in WT plants (Figure 4A). In contrast, compared with WT, the AsA content increased and decreased by 42.38% and 21.96% in OX-P and anti-P, respectively (Figure 4B). The NADP^+^/NADPH ratio was 1.28-fold and 0.35-fold lower and higher in anti-P and OX-P, respectively (Figure 4C). The MDA content increased and decreased by 41.99% and 37.29% in anti-P and OX-P, respectively (Figure 4D). The ADP/ATP ratio decreased by 30.78% in OX-P, but increased by 17.49% in anti-P (Figure 4E). The APX activity increased by 18.05% in OX-P, but no significant change in anti-P compared with WT (Figure 4F). These results showed that the downregulation or upregulation of PtotAPX in transgenic P. tomentosa did not lead to significant changes in photosynthetic efficiency, although it changed the level of H_2_O_2_ and protein oxidation.

Previous research showed that APX plays a crucial role in the removal of ROS from chloroplasts to improve their tolerance to oxidative stress [4]. Thus, we examined the H_2_O_2_ content, reduced AsA, total AsA ratio (reduced AsA/total AsA), and NADP^+^/NADPH ratio under oxidative stress conditions. The H_2_O_2_ content in wild-type leaves treated with MV was 4.45-fold higher than that in the control (Figure 4A). In control and treated plants, it decreased by 55.46% of the reduced AsA/total AsA ratio and increased by 127.91% of NADP^+^/NADPH ratio in anti-*PtotAPX* plants under oxidative stress, respectively (Figure 4B,C). Furthermore, the H_2_O_2_ content and NADP^+^/NADPH ratios decreased by 41.39% and 35.43% in OX-P, whereas they increased by 25.01% of H_2_O_2_ content in anti-P plants compared to WT under MV-treated conditions (Figure 4A,C). In contrast, the reduced AsA/total AsA ratio increased by 48.45% in the OX-P plants. Wild-type plants subjected to MV treatment showed a higher MDA level than the control, at nearly 5.21-fold that of untreated WT plants (Figure 4D). Under oxidative stress conditions, the MDA concentration increased significantly (41.99%) in anti-P compared to wild-type plants subjected to the treatment. However, it showed a lower MDA level in OX-P plants than in WT plants subjected to the same treatment (37.29%) (Figure 4D). In addition, another indicator of the functional status of cells and cell activity, the ADP to ATP ratio (ADP/ATP), increased by 22.63% in oxidative stress-treated wild-type plants compared to the control. Moreover, the ADP/ATP ratio increased and decreased by 42.71% and 26.89% in anti-P and OX-P plants under oxidative stress conditions, respectively, compared with wild-type plants under the same treatment (Figure 4E). These results suggest that high levels of H_2_O_2_ were the cause of oxidative damage to the chloroplasts. The absence of *PtotAPX* increased the accumulation of H_2_O_2_, resulting in damage to the chloroplast structure under oxidative stress.

### 2.4. Enzymatic Properties of Chloroplast PtotAPX

As shown in Figure 4, the AsA ratio significantly decreased under oxidative stress. Therefore, we considered whether the lack of reduced AsA affected the function of *PtotAPX*. The enzyme activity of PtotAPX was determined using the AsA and H_2_O_2_ substrates.

During previous investigations, we obtained the full-length sequence of the cDNA of *PtotAPX*. The expression construct *pET30a-PtotAPX* was expressed in *Escherichia coli*, resulting in the production of recombinant PtotAPX. To analyze the enzymatic properties of PtotAPX, purified recombinant proteins were used for enzyme activity assays. The optimum pH of PtotAPX is 7.4–8.0. However, the activity is relatively insensitive to pH changes, as it can retain more than 70% activity at pHs 5.8 and 9.2 (Appendix A). The activity of PtotAPX is highest at 20 °C and changes most significantly between 8 °C and 16 °C (Appendix A). The concentration of one of the substrates, AsA and H_2_O_2_, was fixed to a relative excess, and the activity of the enzyme was measured under different concentration gradients of the other substrate (Table 1). The measured reaction rate followed Michaelis–Menten-type kinetics. At a fixed H_2_O_2_ concentration, the *K*_m_ and *V*_max_ values of PtotAPX for AsA were 0.66 ± 0.27 mM and 25.12 ± 5.98 mM min^−1^ mg^−1^, respectively. At a fixed AsA concentration, the *K*_m_ and *V*_max_ values of PtotAPX for H_2_O_2_ were 0.02 ± 0.002 mM and 14.77 ± 0.25 mM min^−1^ mg^−1^, respectively. Taken together, these results indicate that PtotAPX could utilize lower concentrations of AsA to scavenge H_2_O_2_. The decrease in the reduced AsA content could reduce the removal of high concentrations of H_2_O_2_ with PtotAPX.

We displayed the changes present in the three plants in response to treatment. However, a straightforward relationship with the presence and activity of enzymes is still not clear. To explain the differences between the presence and absence of enzymes, as well as activity between the three types of plants, an enzyme activity experiment was provided. The activity of APX in anti-PtotAPX has no significant difference compared with WT plants under normal growth conditions, but a nearly 1.18-fold increase was observed in OX-PtotAPX compared with WT (Figure 4F). When plants were treated with MV for 5 h, the activities in both WT and transgenic plants remarkably decreased. The APX activity in OX-PtotAPX was 1.32-fold that in WT plants. However, it decreased by 0.63-fold in anti-PtotAPX compared to WT plants (Figure 4F). These results suggest that the presence of PtotAPX could change the activity of the enzyme, which might be the reason for the slight or obvious differences in oxidative and reduced contents between the three types of plants when untreated or treated with MV, respectively. Overexpressing PtotAPX can improve its oxidative stress tolerance by increasing the activity of antioxidant enzymes.

## 3. Discussion

### 3.1. Chloroplast PtotAPX, Acting as One of the Components in Antioxidant Defense Systems Genes, Can Be Complemented by Other Antioxidant Defense Systems

Several studies have demonstrated that ROS can act as important signaling molecule, as well as toxic molecules, which can cause damage to cells [17]. Thus, the concentration of H_2_O_2_ should be tightly regulated [18]. The level of H_2_O_2_ was maintained at a certain level under normal conditions. Our results also showed that, under untreated growth conditions, the phenotype of plants and subcellular structure of chloroplasts were not significantly different between the wild-type and transgenic plants (Figure 1A–I). Previous studies also suggested that chloroplast knockdown lines, as well as single and double null mutants in chloroplast APX (chlAPX; sAPX and tAPX) in *Arabidopsis* and rice, exhibited a normal phenotype and normal biochemical and physiological attributes under normal growth conditions [19].

However, it remained unclear why wild-type and transgenic plants exhibited minor visible phenotypic differences under untreated conditions. Previous studies have shown that low levels of H_2_O_2_ cannot cause oxidative damage to plants. Although the levels of H_2_O_2_ and redox states in the chloroplasts of anti-*PtotAPX* plants slightly increased (Figure 4A), they had no effect on chloroplast structure and plant growth (Figure 1D–I). A *tapx*-silence mutant in *Arabidopsis* appeared to exert a minor influence under normal light conditions, in which chloroplastic H_2_O_2_ levels increased [20]. Normally, xylem cell wall development is closely related to plant resistance to stress [21]. The levels of H_2_O_2_ affect lignin polymerization in response to stress. Our results showed that the size of fiber cells of wild-type and transgenic plants was not significantly different. Although the thickness of the fiber cell wall in anti-P plants was clearly less than that of the wild-type plants, it was greater in the OX-P plants, suggesting that the wild-type and transgenic plants had no significant difference in the thickness of their stems (Figure 2). This indicated that the silencing of tAPX had no influence on chloroplasts, and the increased H_2_O_2_ levels in chloroplasts were not high enough to affect plant growth (i.e., not cytotoxic) [8].

Moreover, research has shown that, during plant growth, other antioxidant enzymes act together to scavenge H_2_O_2_ in cells, such as CAT, SOD, and GPX [22,23,24]. Compensation by other antioxidant enzymes can scavenge the remaining H_2_O_2_ in chloroplasts. Thus, the chloroplast peroxidase, PtotAPX, might be one of the components of the antioxidant system, providing a reasonable explanation for the negligible phenotypic difference between anti-*PtotAPX* and OX-*PtotAPX* plants compared to wild-type plants. It has also been proposed that cytosolic APX (cAPX) mediates cross-compartment protection in chloroplasts when the chloroplast APX is missing [25]. Our results showed that the activities of APX had a slight change in transgenic plants compared with WT under normal growth conditions, consistent with the slight change in the ROS level between the three types of plants (Figure 4A–F). Therefore, the antioxidant networks of various cellular compartments and antioxidant enzymes play important roles in regulating the balance of H_2_O_2_ levels.

### 3.2. PtotAPX Is a Key Enzyme in Chloroplasts That Scavenges H_2_O_2_ under Oxidative Stress, Resisting Oxidative Stress Damage

The balance between the production and scavenging of H_2_O_2_ in chloroplasts is one of the key determinants of plant acclimation to stress conditions [8], with H_2_O_2_ levels that are too low or too high impair plant growth [7]. Under normal conditions, the lack of tAPX in *Arabidopsis* and *P. tomentosa* did not affect plant growth. However, when plants were treated with MV, the H_2_O_2_ level reached the threshold boundary, where excessive H_2_O_2_ affected the structure and function of chloroplasts, subsequently influencing plant growth and development (Figure 3). Overexpression tobacco tAPX increased tolerance to oxidative stress caused by MV [6]. Our results also showed that the chloroplast structure of transgenic and wild-type plants was damaged after 5 h of treatment with MV. Moreover, the damage of anti-P leaves was more severe than that of WT under the same conditions. A large number of transparent spots appeared on anti-P leaves, while no significant changes were observed in OX-P leaves compared to the wild-type control (Figure 3A).

Plants produce excessive amounts of ROS under various adverse environmental conditions. Increased H_2_O_2_ levels were observed in the wild-type and anti-P plants, but decreased in the OX-P plants (Figure 4A). The accumulation of ROS often results in direct or indirect oxidative damage in plant cells subjected to a series of stresses and may cause significant damages to cellular, destroying chloroplast structure [26]. Excessive H_2_O_2_ destroyed chloroplast structure and function compared with the WT. Although the membrane system of OX-P chloroplast was intact and its structure was well-developed, the internal structure of anti-P plants was degraded and the membrane system was incomplete (Figure 3D–I). Duan et al. also indicated that stress induces cellular ultrastructural disorders [4]. Thus, when subjected to serious stresses resulting in the content of H_2_O_2_ in the chloroplast exceeding the threshold, the extent of cross-compartment protection might be limited, causing damage to leaves. For example, the lack of tAPX in tobacco (*Nicotiana tabacum*) and wheat (*Triticum aestivum*) adversely affects plant growth and photosynthetic performance under stress conditions [6]. The overexpression of tAPX transgenic tobacco plants (TpTAP-12), whose activity was approximately 37-fold higher than that of the wild-type plants, resulted in no significant differences in various physiological parameters compared to wild-type plants under controlled conditions, but showed an increased tolerance to 50 μM MV stress [6].

Rice tAPX (OsAPX8) RNAi-inhibited transgenic plants were hypersensitive to high salt, but exhibited a normal morphological phenotype under normal or low-salt conditions, indicating that OsAPX8 was the major H_2_O_2_ scavenging enzyme when challenged with a high salt concentration, but could be replaced by other APXs under low-salt or normal conditions [27]. Likewise, tAPX showed different responses under different materials and different levels of stress treatment conditions. It seemed that chlAPXs played an important role in the response to stress under severe stress conditions, and were the main target of the ROS scavenging system in the chloroplast. After treatment with 100 μM MV, the H_2_O_2_ content increased rapidly in anti-P, to nearly 8-fold that of wild-type plants under normal conditions. In addition, we also found that APX activity increased in OX-P-M and H_2_O_2_ content decreased, so that the gene related to lignin synthesis was upregulated, which was consistent with the results of stem TEM. However, excessive H_2_O_2_ accumulated in anti-P-M plants caused oxidative damage and could not be used to polymerize lignin. We speculated that the reduction in *PtotAPX* leads to the decrease in APX enzyme activity and the increase in H_2_O_2_ content in anti-P plants, resulting in resistance to the damage of oxidative stress. Therefore, *PtotAPX* plays a significant role in plant growth to eliminate the accumulation of ROS under oxidative stress under severe stress conditions, reflecting the importance of *PtotAPX* in the antioxidant defense system.

### 3.3. The Absence of AsA Affects the Function of PtotAPX

Since tAPX was localized in the thylakoid membrane, it was directly involved in the water–water cycle and ROS-scavenging [4]. AsA was used as an electron donor by tAPX to reduce H_2_O_2_ back into water, playing an important role as a redox buffer and preventing the oxidation of metabolites [8]. Using AsA as a substrate, the activity of the enzyme PtotAPX was measured under different concentration gradients of other substrates. Compared to the results shown by Yin et al. [28], PtotAPX has a greater affinity for AsA (Table 1). Taken together, these results indicate that PtotAPX could utilize lower concentrations of AsA to scavenge H_2_O_2_.

Our results show that the reduced AsA/total AsA ratio of anti-P plants was not significantly different from that of wild-type plants under untreated conditions (Figure 4B), suggesting that there was sufficient AsA level to reduce H_2_O_2_. However, AsA can be oxidized by ROS by environmental stress, leading to a reduction in the content of reduced AsA. At low AsA levels, tAPX was extremely sensitive to H_2_O_2_. After treatment with MV, the AsA/total AsA ratio of wild-type plants decreased by 42.39% compared to that of untreated wild-type plants. In addition, the AsA/total AsA ratio of anti-P plants under oxidative stress was clearly reduced compared with wild-type plants under the same stress, but increased in OX-P plants. Thus, under oxidative stress conditions, a significant decrease in the reduced AsA content was observed for *anti-P,* leading to the inactivation of tAPX activity, but not in OX-P. High levels of tAPX could prevent the depletion of AsA in the surface region of the thylakoid membranes. AsA content and PtotAPX activity are important for maintaining H_2_O_2_ balance and WWC capacity. Increasing evidence showed that the activities of antioxidative enzymes were associated with tolerance to abiotic stresses. Our results showed that the APX activities in antisense plants had no difference compared to WT, and was slightly higher in OX-P plants, consistent with the Pn. The higher APX activities in OX-P plants under oxidative stress could resist the damage of stress, and the results are consistent with the higher integrity of the structure of chloroplast as well as the downregulated expression of ROS-related genes compared with WT in OX-P plants (Figure 4F). On the contrary, the APX activity of antisense plants was lower, so a high expression of resistance genes is required to respond to oxidative stress (Figure 4F). Thus, PtotAPX plays a positive role in H_2_O_2_ detoxification and requires the sufficient synergy of AsA levels [29].

## 4. Materials and Methods

### 4.1. Plant Materials and Treatments

*Populus tomentosa* was used as a donor for genetic transformation and as the wild-type control. The WT and transgenic plants were grown on solid 1/2 Murashige and Skoog (MS) medium supplied with 1.5% (*w*/*v*) sucrose, 100 mg/L inositol, 0.1mg/L NAA, and 6 g/L agar, pH 5.8, in a growth chamber with a 14 h light/10 h dark photoperiod at 25 °C [15] and grown for two months. Subsequently, the 2-month-old wild-type and transgenic plants were exposed to 100 μM MV for 5 h to induce oxidative stress. The plant leaves were then used for experimental analyses, such as the determination of various physiological indicators and electron microscopy. Three-month-old WT and transgenic plants were transferred into nutrient soil and grown under a 14 h light/10 h dark photoperiod at 25 °C, using for phenotypic observation and Pn and Fv/Fm analyses.

### 4.2. Molecular Cloning and Plasmid Construction

A 1233 bp ORF of *PtotAPX* was amplified by PCR using the primers *PtotAPX*-F/R, and the nucleotide sequence was confirmed by sequencing. Full-length *PtotAPX* cDNA was amplified using OX-*PtotAPX*-F/R and cloned into pBI121 to generate 35S:*PtotAPX*. Full-length *PtotAPX* cDNA was amplified using anti-*PtotAPX*-F/R and cloned into pBI121. The *PtotAPX* coding region without the signal sequence was amplified by PCR with the primers P-*PtotAPX*-F/R and inserted downstream of the pET30a plasmid T7 promoter (Novagen, Madison, WI, USA). The primer sequences are shown in Appendix A.

### 4.3. qRT-PCR

Total RNA was isolated from the WT and transgenic plants using an RNAprep Pure Plant Kit (Tiangen Biotech, Beijing, China) according to the manufacturer’s instructions. First-strand cDNA was synthesized using a FastQuant Reverse Transcription Kit (Tiangen Biotech, Beijing, China). Quantitative reverse transcription PCR (qRT–PCR) was performed using gene-specific primers and SuperReal PreMix Plus (SYBR Green; Tiangen Biotech) on a CFX Maestro software Real-Time System (Bio-Rad, Hercules, CA, USA). The relative expression level of each gene was calculated by the 2^−∆∆Ct^ method, using the ∆Ct of WT as 1. The expression of *PtotAPX* in wild-type and transgenic plants was assessed by qRT-PCR using the qRT-*PtotAPX*-F/R primers described by Li et al. [15]. The *P. tomentasa Actin* gene was used as an internal control for normalization. The primer sequences are shown in Appendix A.

### 4.4. Transmission Electron Microscopy (TEM)

When the wild-type and transgenic *P. tomentosa* plants had grown for 2 months, the second leaves and the first basal stem nodes were collected for TEM. After treatment with 100 μmol m^−2^ s^−1^ MV for 5 h, the leaves were collected for TEM according to the method described by Zhang et al. [30]. Samples were fixed with 2.5% glutaraldehyde in 0.2 M sodium phosphate buffer (pH 7.0), washed with the same buffer, and then post-fixed with 1% osmic acid in phosphate-buffered saline (PBS) for 1 h and washed with PBS. Samples were dehydrated in a graded acetone series, and then embedded in epoxy resin (SPI-Chem™ Low Viscosity Spurr Kits, SPI Supplies, West Chester, PA, USA) according to the manufacturer’s instructions. The plant material was cut into 70 nm ultrathin sections on a Leica UC6 ultramicrotome (Ultracut UCT; Leica; Germany). Sections were examined using an H-7650 transmission electron microscope at 80 kV (Hitachi) and an 832 charge-coupled device camera (TEM; Hitachi, Tokyo, Japan). Fiber cell wall thickness and size were measured using ImageJ software (https://imagej.nih.gov/ij, accessed on 12 March 2022).

### 4.5. Stem Histochemistry

For histological analyses, the first basal stem nodes were collected from the main stem from the wild-type and transgenic plants to generate paraffin sections, according to the procedure described by Han et al. [31]. WT and transgenic plant samples were fixed in formaldehyde/acetic acid/alcohol fixation buffer (FAA; 70% ethanol: formaldehyde: acetic acid = 90:5:5) for 24 h at 4 °C and then stored in 70% ethanol. The stem nodes were gradually dehydrated with 70%, 80%, 95%, and 100% ethanol for 1h in each step. Then, 0.1% safranine O in ethanol was added to pre-stain the xylem, and embedded in paraffin (Sigma-Aldrich). The plant material was cut into 8 µm ultrathin sections on a Leica RM2016 microtome, dewaxed in dimethylbenzene, mounted in neutral balsam after drying, and observed under a Leica DCF500 microscope. Micrographs of sections from 3 specimens were examined. Statistical differences were determined. Xylem thickness and phloem thickness were measured using ImageJ software (https://imagej.nih.gov/ij, accessed on 12 March 2022). Statistical differences were determined using Student’s *t*-test.

### 4.6. Measurement of Net Photosynthetic Rate (Pn) and Maximal Photochemical Efficiency of PSII (Fv/Fm)

The Pn was measured using a portable photosynthetic system (CIRAS-2, PP Systems, Herts, UK) under the following conditions: ambient CO_2_ concentration (360 μL L^−1^), photon flux density (PFD) of 800 μmol m^−2^ s^−1^, and 21% O_2_. Pn was measured after treating the plants under 100 μmol m^−2^ s^−1^ at 25 °C for 30 min. To calculate the Pn of the plants, three pieces of functional leaves from three plants were evaluated to create a standard curve, which was repeated four times at each location. Maximal PSII quantum yield (Fv/Fm) values were measured using Dual-PAM-100 (Walz) after 20 min of dark adaptation. The Fo (minimum fluorescence yield) was measured under weak modulated measuring light (10 µmol photons m^−2^ s^−1^) and the Fm (maximum fluorescence yield) was measured by applying a saturating pulse of white light (4000 mmol photons m^−2^ s^−1^ for 0.8 s). To further validate the signal of chlorophyll fluorescence, the fluorescence of leaves was measured using imaging PAM (instruction manual imaging PAM, Heinz-Walz GmbH, Germany) according to the manufacturer’s instructions, using the same conditions above. The PSII quantum yields (Y(II) = (Fm−F)/Fm) and Fv/Fm (Fv/Fm = (Fm−Fo)/Fm) were reported.

### 4.7. Determination of H_2_O_2_ and Malondialdehyde (MDA) Contents, Reduced AsA/Total AsA, NADP^+^/NADPH, and ADP/ATP Ratios

Chloroplasts were isolated from the leaves of *P. tomentosa* using the Minute™ Chloroplast Isolation Kit CP-011 (Invent Biotechnologies, USA), and the integrity of the chloroplast is greater than 90% according to the manufacturer’s instructions. The MV-treated and -untreated samples from wild-type and transgenic plants were collected. H_2_O_2_ content, MDA content, ADP, and ATP levels were determined as described by Yin et al. [28]. H_2_O_2_ content was detected using Fluorimetric Hydrogen Peroxide Assay Kit (Sigma-Aldrich, USA). MDA content was detected using Lipid Peroxidation (MDA) Assay Kit (Sigma-Aldrich, USA). ADP and ATP levels were detected using a EnzyLight^TM^ ATP assay (BioAssay Systems, USA). The levels of oxidative and reduced AsA were measured using the CAO method, and absorbance was recorded at 265 nm. NADP^+^ and NADPH levels were determined using an Enzy Chrom Assay Kit (BioAssay Systems).

### 4.8. Enzyme-Activity Assays

Expression and purification of the PtotAPX protein were performed according to the procedure described by Zhang et al. [30]. The enzyme activity of purified recombinant PtotAPX was spectrophotometrically assayed. The dependence of enzyme activity on pH and temperature was measured in a pH gradient of 5.8 to 9.2 and a temperature gradient of 8 to 52 °C. The assay mixture was kept at 20 °C and contained 50 mM potassium phosphate buffer at pH 7.0; 1 mM H_2_O_2_; 1 mM EDTA; the analyzed substrate; and the enzyme. The enzyme activity was estimated using AsA and H_2_O_2_ as substrates.

We analyzed the activities of APX (EC 1.11.1.11) by using an APX activity assay kit (Beijing Boxbio Science Technology, Beijing, China). Leaf tissue (0.1 g) was ground in buffer. After centrifugation 13,000× *g* for 10 min at 4 °C, supernatants were used to determine enzyme activity, and then measured by monitoring the decrease in absorbance at 290 nm. APX activity unit: Each g of tissue oxidizes 1 µmol of ascorbic acid per minute as an enzyme activity unit.

## 5. Conclusions

This study revealed the physiological function of chloroplast PtotAPX in the regulation of ROS levels and protection of chloroplasts against oxidative stress in P. *tomentosa*. The data also revealed the differences in chloroplast structure, photosynthesis activity, and oxidized-or-reduced indexes between wild-type and transgenic plants under oxidative stress. The overexpression of *PtotAPX* could maintain ROS dynamic equilibrium under oxidative stress, thus protecting the photosynthetic machinery and efficiency in the chloroplast of *P. tomentosa*. Our results provide a theoretical basis to improve plant abiotic stress through overexpressing chloroplast ascorbate peroxidase in woody plants.

## Figures and Tables

**Figure 1 ijms-23-03340-f001:**
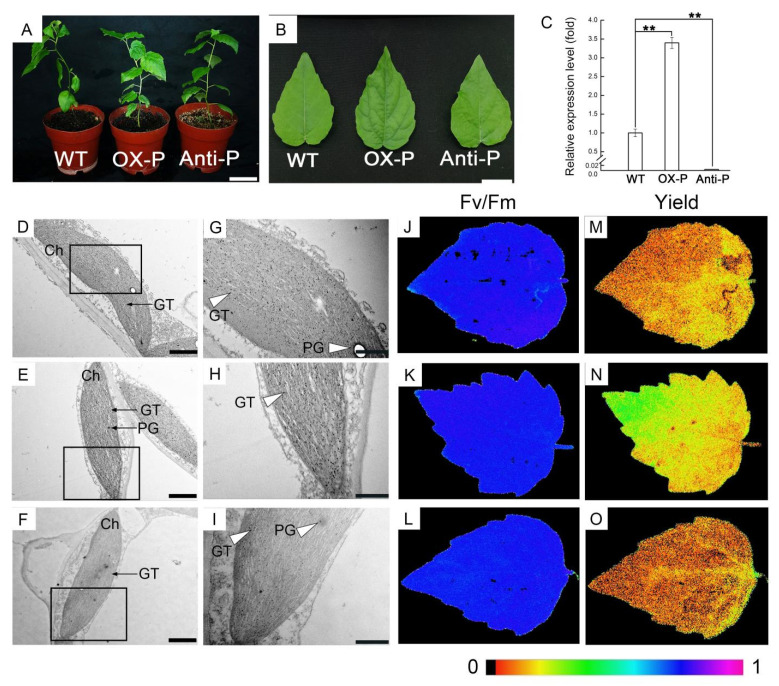
Phenotype, subcellular structure and the change in chloroplast oxidant or reduced contents in transgenic *P. tomentosa*. (**A**,**B**), Growth of transgenic P. tomentosa. (**A**), Bar = 10 cm; (**B**), Bar = 2 cm. (**C**), qRT-PCR of PtotAPX. OX-P, overexpression-transgenic *P. tomentosa*; anti-P, antisense-expression transgenic *P. tomentosa*. Values are means ± SD (*n* = 3). Significant differences between mean values are indicated by asterisks using Student’s *t*-test. ** and * indicate *p* < 0.01 and *p* < 0.05, respectively. (**D**–**I**), Ultrastructural images of chloroplast in leaves. (**D**) and (**G**), WT; (**E**) and (**H**), OX-P; (**F**) and (**I**), anti-P. GT, grana thylakoid; Ch, chloroplast; PG, plastoglobule. (**D**–**F**), Bar = 1 μm; (**G**–**I**), Bar = 500 nm. (**G**–**I**), white arrowheads mark the partial enlarged views of thylakoids in WT, OX-P and anti-P (left), respectively. (**J**–**O**), PSII quantum yield and maximum efficiency of PSII (Fv/Fm). The false color scale is given on the bottom. (**J**–**L**), maximum efficiency of PSII (Fv/Fm). (**M**–**O**), PSII quantum yield. (**J**) and (**M**), WT; (**K**) and (**N**), OX-P; (**L**) and (**O**), anti-P. The false color means the spatial distribution of Fv/Fm values and yield ranged from black (0.0) to purple (1.0).

**Figure 2 ijms-23-03340-f002:**
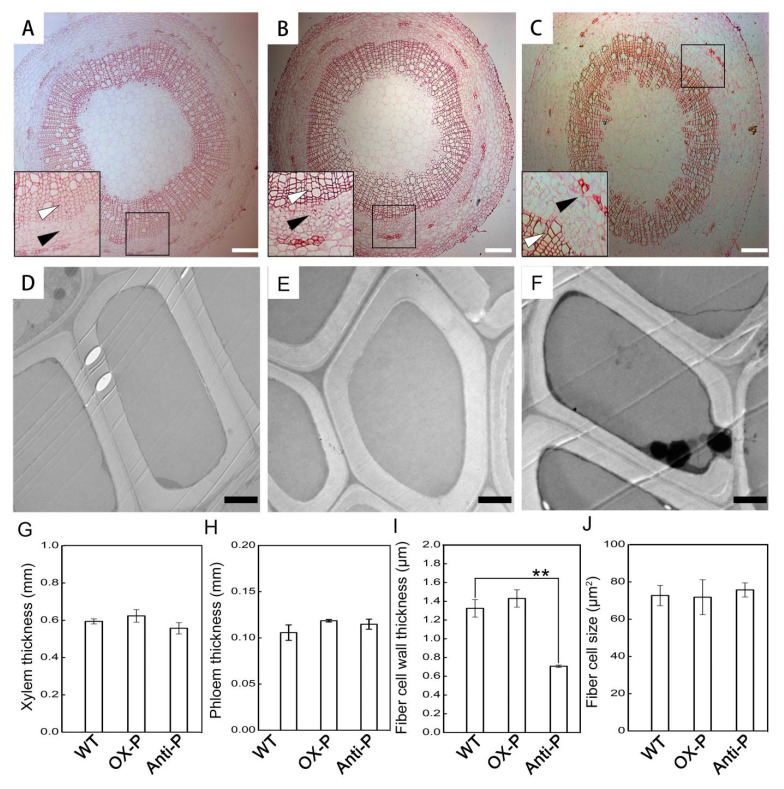
Changes in stems of transgenic plants of *P**. tomentosa*. (**A**–**C**), Paraffin sections in 2-month old plants. (**A**), WT; (**B**), OX-P; (**C**), Anti-P. Bar = 200 μm. (**D**–**F**), Transmission electron micrographs stem transverse sections in 2-month old plants. (**D**), WT; (**E**), OX-P; (**F**), anti-P. Bar = 2 μm. (**G**), Xylem thickness in transgenic plants of WT, OX-P and anti-P during plant growth. Micrographs of sections from 3 specimens were examined. Statistical differences were determined. Values are means ± SD (*n* = 3). (**H**), Phloem thickness in transgenic plants of WT, OX-P and anti-P during plant growth. Micrographs of sections from 3 specimens were examined. Statistical differences were determined. Values are means ± SD (*n* = 3). White or black arrowheads mark the partially enlarged views of xylem or phloem in WT, OX-P and anti-P, respectively. (**I**–**J**), Fiber cell wall thickness or size in transgenic plants. Micrographs of sections from 3 specimens were examined. Statistical differences were determined. Values are means ± SD (*n* = 60 or 30). Significant differences between mean values are indicated by asterisks using Student’s *t*-test. ** and * indicate *p* < 0.01 and *p* < 0.05, respectively.

**Figure 3 ijms-23-03340-f003:**
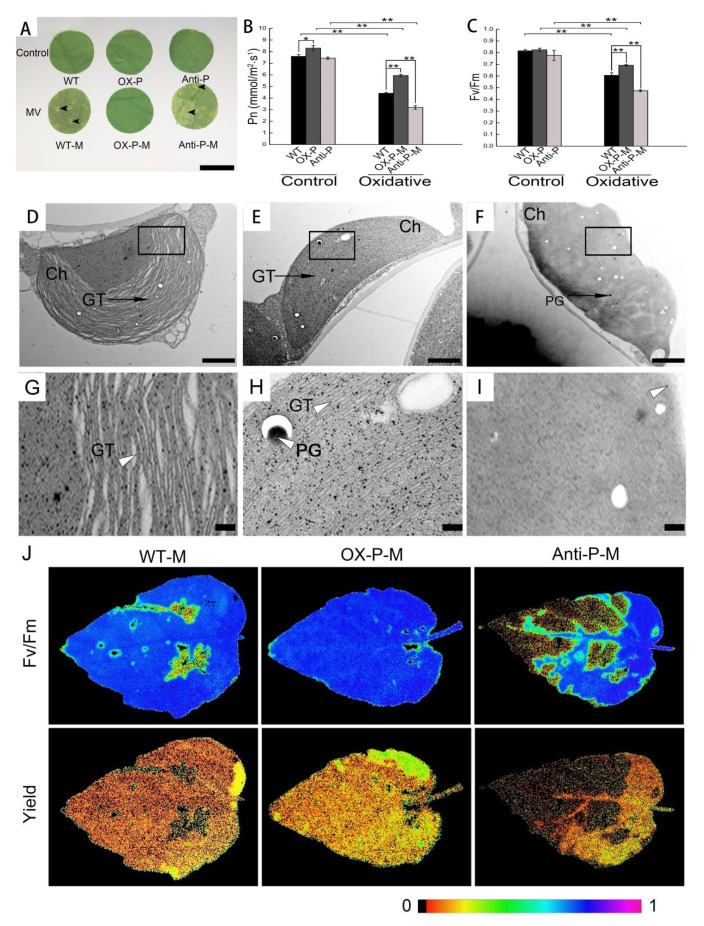
The change in chloroplast oxidant contents and subcellular structure in transgenic *P. tomentosa* under oxidative conditions. (**A**), The change in wild-type and transgenic leaves treated with 100 μM MV for 5 h. WT-P-M, WT treated with MV; OX-P-M, OX-P treated with MV; anti-P-M, anti-P treated with MV. Bar = 0.5 cm. Black arrow-heads in (**A**) marked the transparent spots in transgenic plants treated with methyl viologen (MV). (**B**), Measurement of Pn under 100 μM MV for 24 h. (**C**), Fv/Fm under 100 μM MV for 24 h. (**D**–**I**), Ultrastructural images of chloroplast in leaves under MV oxidative condition. (**D**) and (**G**), WT-M; (**E**) and (**H**), OX-P-M; (**F**) and (**I**), anti-P-M. GT, grana thylakoid; Ch, chloroplast; PG, plastoglobule. Bar = 1 μm. In (**D**–**I**), white arrow-heads marked the partial enlarged views of thylakoids or blurry chloroplast membrane under oxidative conditions in WT-M, OX-P m and anti-P-M, respectively. (**J**), PSII quantum yield of PSII and maximum efficiency of PSII (Fv/Fm) treated with 100 μM MV for 24 h. The false color scale is given on the bottom. The false color means the spatial distribution of Fv/Fm and yield values ranging from black (0.0) to purple (1.0). All values are means ± SD (*n* = 3). Significant differences between mean values are indicated by asterisks using Student’s *t*-test. ** and * indicate *p* < 0.01 and *p* < 0.05, respectively.

**Figure 4 ijms-23-03340-f004:**
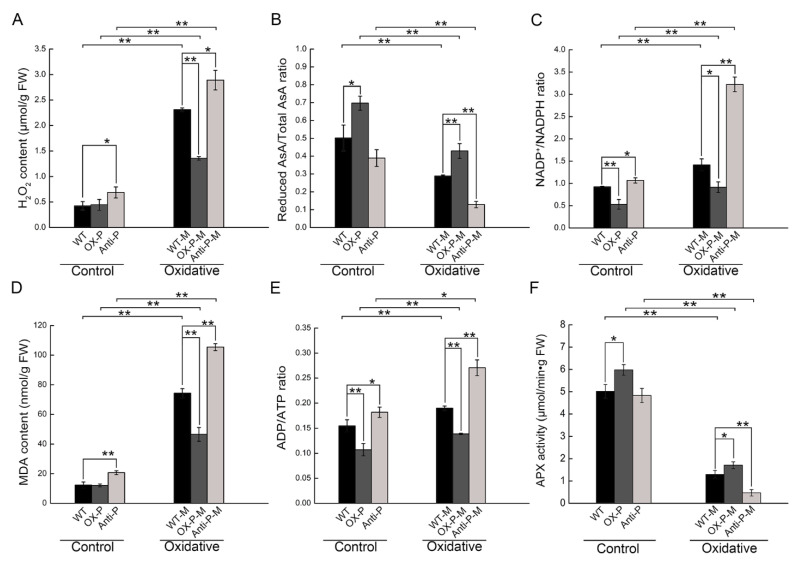
The change in chloroplast oxidant or reduced contents in leaves under 100 μM MV for 5 h. (**A**), Chloroplast H_2_O_2_ content; (**B**), Reduced AsA/total AsA ratio; (**C**), NADP^+^/NADPH ratio; (**D**), Chloroplast MDA content; (**E**), ADP/ATP ratio; (**F**) APX activity. WT-P-M, WT treated with MV; OX-P-M, OX-P treated with MV; anti-P-M, anti-P treated with MV. All values are means ± SD (*n* = 3). Significant differences between mean values are indicated by asterisks using Student’s *t*-test. ** and * indicate *p* < 0.01 and *p* < 0.05, respectively.

**Table 1 ijms-23-03340-t001:** Enzymatic properties of recombinant PtotAPX.

APX Isoforms	Substrate	*K*_m_ (mM)	*V*_max_ (mM min^–1^ mg^–1^)
PtotAPX	AsA	0.66 ± 0.27	25.12 ± 5.98
	H_2_O_2_	0.02 ± 0.002	14.77 ± 0.25

## Data Availability

Sequence data from this article can be found in the NCBI database under accession numbers MT041690 (*PtotAPX*). The data supporting the findings of this study are available from the corresponding author, Hai Lu, upon request.

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
