# Peer review of "Chloroplast Thylakoidal Ascorbate Peroxidase, PtotAPX, Has Enhanced Resistance to Oxidative Stress in Populus tomentosa"

_ijms, 2022, doi:10.3390/ijms23063340_

Round 1
Reviewer 1 Report
The discussed manuscript “Chloroplast thylakoidal ascorbate peroxidase, PtotAPX, enhanced resistance to oxidative stress in Populus tomentosa” is a work containing a lot of important information, but before publishing it, I suggest introducing a few changes.
- The introduction contains a lot of basic, generally known information, but some important data are missing. There is no information there why the authors chose Populus tomentosa for their research.
- Why were the plants treated with methyl viologen? Where did this choice come from. There is no information on this at work
- Lack of a clearly defined research objective
- The manuscript also lacks a summary that would collect all the results together.
Author Response
Response to Reviewer 1 Comments
Dear reviewer:
Thank you very much for your review of our manuscript, entitled “Chloroplast thylakoidal ascorbate peroxidase, PtotAPX, enhanced resistance to oxidative stress in Populus tomentosa”, which we submitted to International Journal of Molecular Sciences. Our specific responses to individual comments are shown below each comment. The revisions made are as follows:
Point 1: The introduction contains a lot of basic, generally known information, but some important data are missing. There is no information there why the authors chose Populus tomentosa for their research.
Response 1: Populus tomentosa is one of the most adaptable tree species in the world, and suitable for planting in drought and high salinity regions. A better understanding of the role about chlAPXs in woody plant could contribute to a more complete comprehension of the molecular and physiological mechanisms to response to abiotic stress. Thus, we chose Populus tomentosa for our research.
Point 2: Why were the plants treated with methyl viologen? Where did this choice come from. There is no information on this at work
Response 2: Currently, oxidative stress is commonly used as an endogenous inducer with MV. MV transfered photosystem I electrons to O2, causing oxidative damage in cells. And then rapidly oxidized by oxygen molecular and reduced oxygen to superoxide, O2-, and high toxic OH radicals finally. It can simulate the situation of photooxidative stress in plants. So we treated the plants with MV to simulate oxidative stress.
Point 3: Lack of a clearly defined research objective
Response 3: The research objective has been added in introduction.
Point 4: The manuscript also lacks a summary that would collect all the results together.
Response 4: The summary has been added.
Reviewer 2 Report
The article manuscript entitled with “Chloroplast thylakoidal ascorbate peroxidase, PtotAPX, enhanced resistance to oxidative stress in Populus tomentosa” was investigated by Li et al. The authors studied the functional role of thylakoid ascorbate peroxidase (tAPX) of wood plant in response to oxidative stress by generating mutants including overexpressor, (OX-PtotAPX), and antisense, anti-PtotAPX. The results suggest that PtotAPX plays a crucial role in maintaining photosynthetic apparatus via analysis of H2O2 levels, MDA contents, and AsA levels as well as of photosynthetic parameters, such as photosynthetic rate (Pn) and PSII (Fv/Fm and Yields). Although it seems like the authors conferred solid sentences in accordance with their results and notions in the manuscript, the manuscript should be appropriately improved as follows below.
- In the figure 1(D-I), it look like the chloroplast sizes in TEM analysis are different. The WT looked the bigger than OX- and anti-PtotAPX although scale bars are similar. How was different in the overall chloroplast sizes among the plants? Please clarify it.
- In the figure 1(J-O), please add the values of Fv/Fm and yield in the picture with statistical analysis.
- In line 128, the explanation of Figure 2B-C does not match with the Figures. Please rectify it.
- What do “False color scale is given on the right” mean in Figure 1 and 3? Is the scale scale at the bottom of the picture?
- In line 131-141, it would be better to move the paragraph to “2.3 decreased PtotAPX content…”. Please reconstruct and rewrite the paragraph.
- In line 140, “of” should not be an italic letter.
- In the figure 2, please add the result of phloem for paraffin section and TEM analysis in the section.
- In the figure 3, the chloroplast of WT was more damaged than OX-PtoAPX under MV treatment in figure 3A. However, in ultrastructural image of chloroplast in leaves under MV, figure E-H looked (OX) more damaged or abnormal than those in figure D-G (WT). Please clarify and replace the image with better resolution.
- In the figure 3J, please add the values of Fv/Fm and yield in the picture with statistical analysis as well.
- In line 184-190, “figure 3E and H”, “figure 3F and I”, “figure 3D-I” did not match with the explanation. Please rectify them.
- In line 270 and 272, the unit style needs to be correct.
- In the materials and methods section, please add more explanation of Dual-pam-100 (Walz) and imaging PAM set conditions (i.e. pulse, blue measuring beam, saturating light flash etc.)
- Please add conclusion section.
- In the figure 1 legend and reference, please check the format style of the Journal. The style is not uniformed and identical with journal recommendation.
Author Response
Response to Reviewer 2 Comments
Dear reviewer:
Thank you very much for your review of our manuscript, entitled “Chloroplast thylakoidal ascorbate peroxidase, PtotAPX, enhanced resistance to oxidative stress in Populus tomentosa”, which we submitted to International Journal of Molecular Sciences. Our specific responses to individual comments are shown below each comment. The revisions made are as follows:
Point 1: In the figure 1(D-I), it look like the chloroplast sizes in TEM analysis are different. The WT looked the bigger than OX- and anti-PtotAPX although scale bars are similar. How was different in the overall chloroplast sizes among the plants? Please clarify it.
Response 1: There was no different in the overall chloroplast sizes among the plants in statistical analysis. Regarding the chloroplast picture of WT in the figure 1D-I, we have replaced it with a more representative one.
Point 2: In the figure 1(J-O), please add the values of Fv/Fm and yield in the picture with statistical analysis.
Response 2: Maximal PSII quantum yield (Fv/Fm) values were measured using Dual-PAM-100 (Walz) and the imageing fluorescence of leaves was measured using imaging PAM (instruction manual imaging PAM, Heinz-Walz GmbH, Germany). So we showed the values of Fv/Fm in Figure 3C only.
Point 3: In line 128, the explanation of Figure 2B-C does not match with the Figures. Please rectify it.
Response 3: In Line 128, we change the Figure 2B-C to Figure 3B-C.
Point 4: What do “False color scale is given on the right” mean in Figure 1 and 3? Is the scale scale at the bottom of the picture?
Response 4: We have changed the “False color scale is given on the right” to “False color scale is given on the bottom”.
Point 5: In line 131-141, it would be better to move the paragraph to “2.3 decreased PtotAPX content…”. Please reconstruct and rewrite the paragraph.
Response 5: The description of Pn and Fv/Fm as well as chloroplast oxidant or reduced contents under normal growth conditions have been moved to “2.3 decreased PtotAPX content…”.
Point 6: In line 140, “of” should not be an italic letter.
Response 6: The “of” is corrected.
Point 7: In the figure 2, please add the result of phloem for paraffin section and TEM analysis in the section.
Response 7: The paraffin section in Figure 2A-C showed the xylem and phloem. We also supplemented the statistical analysis of phloem thickness.
Point 8: In the figure 3, the chloroplast of WT was more damaged than OX-PtoAPX under MV treatment in figure 3A. However, in ultrastructural image of chloroplast in leaves under MV, figure E-H looked (OX) more damaged or abnormal than those in figure D-G (WT). Please clarify and replace the image with better resolution.
Response 8: After a 5-h oxidative stress treatment, compared with the untreated control, the thylakoid lamellar structure of WT began to become loose and disordered, and the structure of the chloroplasts seemed to be more globular. The distance between the pairs of grana membranes is one of the indicators to detect chloroplast development and damage. The distance between the pairs of grana membranes in WT treated with MV was larger than that in OX. Thus, the chloroplast was more damaged than WT.
Point 9: In the figure 3J, please add the values of Fv/Fm and yield in the picture with statistical analysis as well.
Response 9: Maximal PSII quantum yield (Fv/Fm) values were measured using Dual-PAM-100 (Walz) and the imageing fluorescence of leaves was measured using imaging PAM (instruction manual imaging PAM, Heinz-Walz GmbH, Germany). So we showed the values of Fv/Fm in Figure 3C only.
Point 10: In line 184-190, “figure 3E and H”, “figure 3F and I”, “figure 3D-I” did not match with the explanation. Please rectify them.
Response 10: In line 184-190, “figure 3E and H”, “figure 3F and I”, “figure 3D-I” have corrected, which “figure 3E and H” are OX-PtotAPX and “figure 3F and I” are anti-PtotAPX.
Point 11: In line 270 and 272, the unit style needs to be correct.
Response 11: In line 270 and 272, the unit style was changed to mM and mM min-1 mg-1.
Point 12: In the materials and methods section, please add more explanation of Dual-pam-100 (Walz) and imaging PAM set conditions (i.e. pulse, blue measuring beam, saturating light flash etc.)
Response 12: The set conditions have been added of Dual-pam-100 (Walz) and imaging PAM in the materials and methods section.
Point 13: Please add conclusion section.
Response 13: The conclusion section has been added.
Point 14: In the figure 1 legend and reference, please check the format style of the Journal. The style is not uniformed and identical with journal recommendation.
Response 14: The style of figure legend and reference has been changed and is identical with the journal.
Round 2
Reviewer 1 Report
Accept in present form
Author Response
Thank you very much for your review of our manuscript.
Reviewer 2 Report
Thank the authors for addressing all of my points and concerns. However, the font style in the figure 1 and reference does not still match with others in the manuscript. Besides, the conclusion section needs to be extensively improved.
Author Response
Dear reviewer:
Thank you very much for your review of our manuscript, entitled “Chloroplast thylakoidal ascorbate peroxidase, PtotAPX, enhanced resistance to oxidative stress in Populus tomentosa”, which we submitted to International Journal of Molecular Sciences. Our specific responses to individual comments are shown below each comment. The revisions made are as follows:
Point 1: Thank the authors for addressing all of my points and concerns. However, the font style in the figure 1 and reference does not still match with others in the manuscript. Besides, the conclusion section needs to be extensively improved.
Response 1: Thanks again for your careful correction. The font style in the figure 1 and references have altered. And the conclusion section has been revised and improved